# Ex Vivo Vascular Imaging and Perfusion Studies of Normal Kidney and Tumor Vasculature

**DOI:** 10.3390/cancers16101939

**Published:** 2024-05-20

**Authors:** Ragnar Hultborn, Lilian Weiss, Egil Tveit, Stefan Lange, Eva Jennische, Malin C. Erlandsson, Martin E. Johansson

**Affiliations:** 1Department of Oncology, Institute of Clinical Sciences, Sahlgrenska Academy, University of Gothenburg, 405 30 Gothenburg, Sweden; ragnar.hultborn@oncology.gu.se; 2Department of Physiology, Institute of Neuroscience and Physiology, Sahlgrenska Academy, University of Gothenburg, 405 30 Gothenburg, Sweden; lcweiss@hotmail.com; 3Department of Surgery, Sahlgrenska University Hospital, 413 45 Gothenburg, Sweden; egil.tveit@gmail.com; 4Department of Microbiology, Institute of Biomedicine, Sahlgrenska Academy, University of Gothenburg, 405 30 Gothenburg, Sweden; stefan.lange@microbio.gu.se; 5Department of Medical Chemistry and Cell Biology, Institute of Biomedicine, Sahlgrenska Academy, University of Gothenburg, 405 30 Gothenburg, Sweden; eva.jennische@anatcell.gu.se; 6Department of Rheumatology and Inflammation Research, Institute of Medicine, Sahlgrenska Academy, University of Gothenburg, 405 30 Gothenburg, Sweden; malin.erlandsson@rheuma.gu.se; 7Department of Laboratory Medicine, Institute of Biomedicine, Sahlgrenska Center for Cancer Research, Sahlgrenska Academy, University of Gothenburg, 405 30 Gothenburg, Sweden; 8Department of Clinical Pathology, Sahlgrenska University Hospital, 413 45 Gothenburg, Sweden

**Keywords:** renal cell carcinoma, renal vasculature, barium sulphate contrast, microangiography, autoradiography, micro-CT, Immunohistochemistry, dark-field microscopy

## Abstract

**Simple Summary:**

Organs as well as cancer require a supply of nutrients and oxygen and removal of waste products. These tasks are carried out by the vascular system. Knowledge of the vascular properties in organs and tumors is key for understanding normal and abnormal function. For cancer, vascular function is also highly relevant to understand response to treatment, metastasis, and tumor progression. In this study, we use various techniques to characterize the vascular tree and flow in kidneys with kidney cancer. We connected kidneys to a perfusion system and used barium sulphate contrast to visualize the vascular architecture contact microangiography. Immunohistochemistry was used to visualize the vessels in relation to perfusion. The vascular resistance was measured using the radioactive microspheres and in cases that were feasible, we used micro-CT to characterize the vascular tree. This work aims to suggest the use of these techniques for any organ or tumor available for ex vivo perfusion.

**Abstract:**

This work describes a comprehensive study of the vascular tree and perfusion characteristics of normal kidney and renal cell carcinoma. Methods: Nephrectomy specimens were perfused ex-vivo, and the regional blood flow was determined by infusion of radioactive microspheres. The vascular architecture was characterized by micronized barium sulphate infusion. Kidneys were subsequently sagitally sectioned, and autoradiograms were obtained to show the perfusate flow in relation to adjacent contact X-ray angiograms. Vascular resistance in defined tissue compartments was quantified, and finally, the tumor vasculature was 3D reconstructed via the micro-CT technique. Results show that the vascular tree of the kidney could be distinctly defined, and autoradiograms disclosed a high cortical flow. The peripheral resistance unit of the whole perfused specimen was 0.78 ± 0.40 (*n* = 26), while that of the renal cortex was 0.17 ± 0.07 (*n* = 15 with 114 samples). Micro-CT images from both cortex and medulla defined the vascular architecture. Angiograms from the renal tumors demonstrated a significant vascular heterogeneity within and between different tumors. A dense and irregular capillary network characterized peripheral tumor areas, whereas central parts of the tumors were less vascularized. Despite the dense capillarity, low perfusion through vessels with a diameter below 15 µm was seen on the autoradiograms. We conclude that micronized barium sulphate infusion may be used to demonstrate the vascular architecture in a complex organ. The vascular resistance was low, with little variation in the cortex of the normal kidney. Tumor tissue showed a considerable vascular structural heterogeneity with low perfusion through the peripheral nutritive capillaries and very poor perfusion of the central tumor, indicating intratumoral pressure exceeding the perfusion pressure. The merits and shortcomings of the various techniques used are discussed.

## 1. Introduction

All tissues require a constant supply of nutrients and oxygen with concomitant waste disposal by the vascular supply. Tumors also require these functions for growth and dissemination in case of malignancy. The physiological properties of organs, such as the kidney or liver, depend on a structurally highly organized vascular architecture [1,2,3]. Furthermore, the study of the vascular tree of malignancies has, during recent decades, received increased attention, since the ever-expanding therapeutic arsenal ultimately depends on vascular access to the tissue. Today, anti-angiogenesis by receptor tyrosine kinase inhibition directly targets tumor vasculature, and cytostatics and radiation treatment protocols highly depend on vascular integrity [4,5].

Another valid example is immune checkpoint inhibition, which strongly depends on vascular factors for efficacy. Vascular structure and organization have traditionally been studied via light and electron microscopy. The three-dimensional vascular architecture has been studied via, for example, resin vascular corrosion casts [2,6,7] and, more recently, via confocal microscopy and micro-CT [8,9,10]. The dynamics of blood flow distribution in specific parts of a complex organ, like the kidney, has been studied, e.g., via the labeled microsphere technique [11,12,13]. However, most studies have been performed in animals, and studies on human material are rare, as are studies combining flow measurements with structural investigation. In this study, we used an ex vivo perfusion protocol of human kidneys nephrectomized due to tumors, combined with a variety of morphological techniques and with flow physiological measurements to characterize the vascular tree of both normal and cancerous kidney tissue. We finally discuss the strengths and limitations of the different techniques.

## 2. Materials and Methods

### 2.1. Ethics

The Swedish Ethics Authority approved the study (Dnr 2021-00477).

### 2.2. Procedures

We performed the perfusion study from 1979 to 1984. The kidneys were immediately taken care of in the operating room following a nephrectomy due to a diagnosed renal tumor. The renal vessels were divided, and the artery was cannulated. Subsequently, the kidney was flushed with 5% low-molecular-weight dextran (Perfadex; Pharmacia, Uppsala, Sweden, presently RescueFlow^®^; Øresund Pharma ApS, Copenhagen, Denmark) at room temperature until the venous effluent was clear. The vessels were connected to a perfusion system schematically depicted in Figure 1 [14]. The set-up allowed for the injection of labelled microspheres, 15 µm in diameter as well as withdrawal of reference perfusate volumes, enabling the quantitative estimation of perfusate flow.

### 2.3. Perfusion Technique

Via an arterial cannula, the nephrectomies were connected to rubber tubing (Figure 1) and a peristaltic constant flow pump (Ismatec MP4, Ismatec SA, Glattbrugg, Switzerland). The specimens were perfused at 37 °C with oxygenated perfusate consisting of 4% dextran (70 kD; Macrodex; Pharmacia, Uppsala, Sweden) and 100 mL of horse serum in 1 L of a salt solution with 143 mM Na^+^, 4.3 mM K^+^, 2.5 mM Ca^2+^, 0.83 mM Mg^2+^, 141 mM Cl^−^, 13.3 mM HCO_3_^−^, 0.46 mM H_2_PO_4_ and 5.6 mM glucose. The preparation was immersed in perfusate to prevent gravitational pressure artifacts. The venous effluent was drained into the perfusate. The afferent tubing also accommodated two thin catheters (PE 50) introduced via a T-connection, ending approximately 1 cm within the renal artery. One catheter was used to draw a reference perfusate sample during microsphere injection, while the other was used for pressure recording. Papaverine was given as a bolus dose (median dose 120 mg; range 80 to 160 mg) to induce maximal vascular relaxation. The pump flow was gradually increased to produce a pressure of approximately 30 to 40 mm Hg in the renal artery, and the flow was set at that rate for the rest of the experiment. Microspheres (New England Nuclear, Boston, MA, USA; 15 µm) labeled with I^125^, approximately 300,000 spheres in 1 mL of saline with Tween, were injected with a fine needle through the latex tubing for 45 s to avoid sphere aggregation. Fifteen seconds before, during and 30 s after the microsphere injection, a reference sample was drawn at 2 mL/min (Model 351 Sage infusion pump). Vascular reactivity was determined after norepinephrine injection using another two sets of differently labeled 15 µm microspheres [14]. This part of the experiments will not be further described. Most experiments were concluded with manual perfusion at a higher pressure of a micronized (<1 µm) barium sulphate suspension (Barosperse; Mallinckrodt, Inc., St. Louis, MO, USA) in perfusate medium (0.45 g/mL) with 10 percent formaldehyde [15].

### 2.4. Post-Experimental Handling of the Perfused Kidneys

The kidneys were immersed in formaldehyde solution for a few days, after which perirenal tissue was removed, and the kidney with the tumor was sliced sagitally into approximately 1 to 4 mm thick slices using a kitchen food slicer. The slices were prepared for further experiments in the following way: (a) the thinnest section, 1 mm, was used for contact X-ray microangiography. The sections were placed between plastic films and applied to the envelopes of a Kodak X-omatic film (VWR-Avantor, Kista, Sweden). Exposure was made at 27 kV, 63 mA in a CGR mammograph with conventional film development. The result provided information on the vascular architecture and density and whether any part of the specimen had been inadequately perfused. (b) A slightly thicker section, 2 mm, was used for autoradiography and placed between envelopes of Kodak X-omatic films on either side of the section. One film was exposed for 2 to 3 weeks and, depending upon the degree of blackening, the other was exposed for up to 2 to 3 months. The result provided information on the gross perfusate flow distribution of the microspheres. (c) A thicker section, approximately 3 to 4 mm, was used to take 0.1 to 1 g pieces for quantitative radioactivity measurement to provide perfusate flow data. Five to 20 pieces were taken from cortical and medullary tissues, and 10 to 20 pieces from tumor tissue; each sample was weighed and placed in a tube with 4% formaldehyde. After the radioactivity measurements, these pieces were prepared for routine histology and immunohistochemistry (CD31 for endothelium) to ensure tissue representation and make possible correlation between morphology and physiology. The 4 µm paraffin sections with a surface area of 0.5–1 cm^2^, were macrophotographed at low-angle illumination from below providing a dark-field image of the contrast-filled blood vessels. On top of the paraffin sections, a drop of immersion oil was applied to avoid light scattering from the crystalline paraffin. Before deparaffination, the 4 µm sections were studied via dark-field microscopy, see below, to identify contrast-filled vessels. After that, the same sections were processed for CD31 endothelial immunohistochemistry. The fraction of perfused vessels could thus be estimated despite the loss of contrast during the IHC procedure. An ”England finder” (Labtech, Heathfield, UK), an object glass slide etched with a minute X–Y graticule to enable the location of a selected area in the specimen microscopic slide, was used to identify the same area of the section documented with darkfield microscopy before and after CD31 IHC.

### 2.5. Micro-CT Analysis

The paraffin blocks were scanned in a preclinical micro-CT scanner, SkyScan 1176 micro-CT (Bruker, Antwerp, Belgium). The scanning was conducted at 55 kV and 455 mA with a 0.2 mm aluminum filter. The exposure time was 815 ms. The X-ray projections were obtained at 0.36° intervals with a scanning angular rotation of 180° at a resolution of 9 µm. Data were reconstructed into a three-dimensional (3D) structure using NRECON software (version 1.6.9.8; Bruker) with a beam hardening correction of 30%. The projection images were visualized as three-dimensional images using the software CTVox (version 2.7, Bruker) [16].

### 2.6. CD31 Immunohistochemistry

Four micrometer-thick sections of the paraffin-embedded samples were cut and mounted on Superfrost^®^ Plus glasses (Gerhard Menzel GmbH & Co.KG, Braunschweig, Germany). The sections were then deparaffinized, hydrated and treated with 3% hydrogen peroxide to quench endogenous peroxidase activity. Heat-induced antigen retrieval was performed using an EDTA buffer with pH 8.0. The sections were incubated with a rabbit monoclonal antibody against CD 31 (ab182981, Abcam, Cambridge, UK) at 1/4500 dilution, followed by an anti-rabbit Impress-HRP reagent (Vector laboratories, Inc., Newark, CA, USA). The immunoreactions were visualized using a Liquid DAB + substrate chromogen system (Agilent, Santa Clara, CA, USA), resulting in a brown reaction product.

### 2.7. Dark-Field Microscopy

Dark-field microscopy utilizes the scattered light from structures in the object and prevents direct light from reaching the objective, providing bright scattering structures against a dark background. A Zeiss microscope (Standard 16, Carl Zeiss AG, Oberkochen, Germany) with a phase contrast condenser was used. The condenser was set for high power objectives, but the low power 10x objective excluded direct light from the condenser phase ring to pass to the objective. For high power objectives, a dedicated dark-field Zeiss condenser, including use of substage oil immersion, was used. The micronized barium sulphate particles (<1 µm) scatter the light incident to the specimen much more than the soft tissue, augmenting the visibility of vessels with contrast.

The same sections, 4 µm thick, were then stained for CD31 immunohistochemistry, also best visualized using dark-field according to Jennische et al., allowing for analysis of contrast-filled vessels [17]. Dark-field macro photography (Canon EOS700D, Canon-Sweden, Solna, Sweden) of the sections, 0.5–1 cm^2^, was also performed, see above.

### 2.8. Flow Analysis

Capillary, <15 µm, perfusate flow analysis: Only pieces with more than 50 spheres, as indicated from activity measurement, were included in the analyses. Calculations: Blood flow (Q mL × min^−1^ × 100 g^−1^) estimated by measurement of Iodine^125^ in tissue samples was calculated as follows:tissue flow rate/tissue activity = reference flow rate/reference activity

The reference withdrawal flow rate was 2.0 mL × min^−1^. Peripheral resistance units (PRU) were obtained by dividing the perfusate pressure by the flow and weight for each sample (perfusion pressure mmHg/mL perfusate flow/100 g tissue).

Tumor volume was measured on the fresh sections or contact angiograms by multiplying the longest diameter by the perpendicular and the mean of the two using the formula 4/3 × *π* × D^3^/8.

### 2.9. Histopathology

Hematoxylin-eosin stained 4 µm sections of tumor tissue were scanned in a Hamamatsu slide scanner and classified for tumor type, clear cell renal cell carcinoma, (CCRCC) and oncocytoma) and for malignancy grade according to ISUP (International Society for Urological Pathology) grading.

### 2.10. Clinical Outcome

Postnephrectomy survival time and cause of death were correlated to tumor type and grade.

### 2.11. Statistical Analysis

Linear regression analysis was performed using the SPSS software (IBM SPSS Statistics for Windows, version 28.0 IBM Corp. As cut off for significance a *p*-value < 0.05 was used.

## 3. Results

### 3.1. Clinical Characteristics

During the study period, 28 patients at one center, 20 males and eight females with a mean age of 66 years, were nephrectomized due to renal tumors. They were, according to the surgeon and pathologist, radically removed. Preoperative diagnosis was established via angiography, 18 were classified as hypervascularized and the mean diameter of the tumors was approximately 6.5 cm. Distant metastases were known preoperatively in 5 patients. The mean specimen weight was 494 g, and tumor volumes ranged from 8 to 502 cm^3^ with a mean of 164 ± 138 cm^3^ and a median of 127 cm^3^. The experimental procedures did not cause leakage of perfusate to the tumor surface. Histopathological evaluation classified nine tumors as clear cell carcinomas ISUP grade 1, ten tumors as ISUP grade 2, 4 tumors as CCRCC of eosinophilic type, ISUP grade 3, and 3 tumors as CCRCC of sarcomatoid/rhabdomyoid type, ISUP grade 4, Figure 2.

Two were classified as oncocytomas. Survival post nephrectomy ranged from 0 to 35 years, with a median of 9.5 years. Fifteen of the 28 patients succumbed to their tumors. Mean and median survival among these were 3.0 and 2 years, respectively, with a span from 0 to 9 years. Table 1 below summarizes the ISUP grade in specimens H2 to H29 in relation to survival.

The Appendix A detail the individual specimen characteristics, including morphology. Tumor size did not correlate to grade or survival.

### 3.2. Experimental Outcomes

Data on vascular reactivity to norepinephrine has been published previously [14] and will therefore not be discussed further below. Six specimens were not contrast perfused, no angiograms or autoradiograms were acquired. In seven specimens, reference sample withdrawal failed, and regional peripheral resistance could therefore not be calculated, but tumor PRU relative to that in cortical tissue was calculated. Angiography and autoradiography of the sagittal slices could also be studied. In two specimens, the perfusion pressure was not recorded. Thus, neither specimen nor regional PRU could be calculated, but as above, tumor PRU relative to cortical tissue was calculated. Sagittal angio- and autoradiography could be studied.

The mean perfusion pressure at maximal papaverine-induced vascular relaxation was 36 ± 9 mmHg (*n* = 26), resulting in a mean perfusate flow of 55 ± 22 mL/min × 100 g. The peripheral resistance (PRU) of the entire specimens, 0.78 ± 0.40 (mean ± SD), median 0.70 range 0.31–1.59 mmHg/mL perfusate flow/100 g tissue is not based on microsphere trapping, but on the perfusion pressure and flow of individual specimens. The variability is likely due to varying amounts of perirenal tissue and tumor size. An example of an angiogram with a corresponding autoradiogram is shown in Figure 3. The clinical characteristics of the complete material are presented in the Appendix A section.

### 3.3. Normal Renal Tissue

From cortical tissue, 188 samples were taken for analysis. In 115 samples obtained from 22 specimens, the PRU could be determined. The other samples had less than 50 spheres and were not subject to analysis. A cut-off value of 50 spheres was introduced since radioactivity measurements were too low to allow for a statistically adequate calculation below this number. In samples from cortical tissue, specimens not contrast-injected at high pressure at the end of perfusion, the PRU at papaverine relaxation was 0.15 ± 0.06 (*n* = 6 samples). In contrast-perfused ones (with a mean pressure of 208 ± 64 mmHg), it was 0.17 ± 0.07 (*n* = 16 with 109 samples), which was not statistically different from the former. The mean PRU from cortical tissue samples was 0.18, the median 0.17 and range 0.05–0.63 at maximal relaxation (*n* = 22 with 115 samples). From medullary tissue samples with >50 spheres analyzed, the ”PRU” was 1.87 ± 2.3 median 0.83 range 0.44–9.55 (*n* = 17 with 30 samples). This measure is, however, not biologically relevant since the medullary capillary network is secondary to that of the cortical one, where most spheres will be trapped. Furthermore, the driving perfusion pressure is much lower than that of the primary cortical vascular network, making the concept of PRU irrelevant. However, the radioactivity of medullary samples was measured, and a nominal PRU-value could be determined, which indicated that medullary samples had approximately 10% of the numbers of microspheres of the cortical samples; i.e., a significant shunting past the primary capillary network existed. Analysis of the perfusion pressure in relation to the fraction of cortex to medullary PRU (PRUc/m) disclosed an insignificant trend in the displacement of spheres from cortical to medullary tissue at increased pressure, Figure 4.

The vascular network was visualized by identifying contrast-filled vessels as grossly seen in the sagittal contrast angiograms, Figure 3, and in detail in the 4 µm sample sections. In the latter, these vessels are best seen in dark-field illumination, macroscopically, Figure 5 as well as microscopically, Figure 6. The fraction of contrast-filled vessels in relation to all vessels is shown in sections stained for the endothelial marker CD31 on the same sections analyzed for contrast, Figure 6.

Also, dark-field microscopy will enhance the visibility of the peroxidase-stained endothelium [17], as seen in Figure 7 which compares bright and dark-field microscopy. In the Appendix A only dark-field microscopy images are presented for CD31 IHC. In cortical tissue, the glomeruli and peritubular network were contrast-filled to varying degrees (Figure 6A,B).

The filling degree varied between different kidney specimens; see the Appendix A section. In the medullary tissue, frequent contrast-filled vasa recta were seen. However, most CD31-stained vessels were unfilled, Figure 6C,D. Some well-filled vasa recta are likely derived from the efferent vessels from juxtamedullary glomeruli, which pass directly into the medulla without peritubular branching. No preference for contrast-filling in the outer medulla is seen, (Figure 8).

The diameters of various vessels from contrast-infused specimens were compared to those non-infused. Contrast infusion at high pressure widens the vessels in both cortical and medullary tissues, Figure 9. Micro-CT of selected paraffin blocks demonstrated the vascular 3D network seen as arcuate arteries branching into lobular arteries and arterioles afferent to contrast-filled glomeruli, Figure 10. Contrast-filled vasa recta were also seen.

### 3.4. Renal Tumor Tissue

Each specimen included a malignant neoplasm of varying dimensions, as seen in the sagittal sections (Figure 3 and Appendix A). Contrast-infused specimens showed a significant heterogeneity regarding vascular filling and architecture. The tumor periphery was generally densely vascularized, while central tumor areas or parts within them were scarcely vascularized (Figure 3 and Appendix A). In Figure 11, an enlarged area of the periphery of a large tumor and a small one are seen together with corresponding Htx-Eosin-stained sections.

Figure 12A shows the partial contrast fillings of vessels compared to the same area in 12B where vessels are shown by staining for CD31.

There were extreme variations in contrast filling within and between tumors, but consistently higher than that of the cortical tissue of that specimen, see the Appendix A.

Figure 13A shows compression of the microvascular structures, and Figure 13B shows a micro-CT rendering of a great heterogeneity of contrast-filled vessels in analogy with the 2-dimensional contrast angiograms.

## 4. Discussion

This work presents techniques to characterize complex vascular networks that apply to normal kidney or kidney cancer tissue and other tissues/organ, postmortem or excised at surgery [6]. The organ perfusion set-up did not include collection of perfusate from the renal veins and shunting of spheres could therefore not be analyzed. Further, only 15 µm spheres were used, limiting analysis of vascular dimensions. Microscopy of the thin 4 µm sections disclosed few 15 µm spheres, almost exclusively located within glomeruli, which indicated that most spheres were trapped in this primary capillary network. At this section thickness, many spheres are probably lost. We believe this fact does not prohibit us from obtaining a crude estimate of the location of the spheres in the renal vascular tree. However, as measured via sample radioactivity, approximately 10% of the 15 µm spheres reached and were trapped in the medulla. This result could be explained by shunting through arteriovenous passages > 15 µm in the juxtamedullary cortical tissue. McNay et al. found approximately 1% of injected 19 µm spheres in the medulla and 0.5% passing through ex vivo perfused porcine kidneys [12]. The contrast perfusions at relatively high pressures widen the vessels compared to CD31-positive vessels in specimens not perfused by contrast. Some microspheres may have been displaced from the cortical tissue into the medulla, Figure 4. The tumors in our specimens have a high PRU regarding nutritive flow (arteriovenous passages < 15 µm). This tissue might allow an unrecorded volume to pass through arterio-venous passages larger than 15 µm, represented by the densely contrast-filled vasculature in parts of the tumors. Preoperative angiography in some of the patients in this material demonstrated hyper-vascularized tumors corresponding to our findings. Tumors with a rapid contrast passage indicate the presence of arteriovenous shunts. A clinical study comparing a rapid radiographic arterio-venous passage with measuring ^99m^Tc-MAA demonstrated a 15–57% shunting [18]. Also significant shunting has been reported in similar tumors such as hepatic adenocarcinoma (PMID: 2813766).

The angiograms demonstrate a heterogeneous distribution of contrast-filled vessels, which makes a vascular/capillary density or area classification difficult. Most tumors included a dense vascular network and parts devoid of vessels in necrotic areas. Only two small tumors showed a dense network throughout (H11 and H25). Also, these tumors had a high vascular resistance as measured with 15 µm spheres. Thus, it was difficult to classify them into well-defined categories concerning the vascular pattern. In conclusion, the vascular density does not translate into degree of perfusion. Tumor areas with an extreme vascular density are often poorly perfused compared to cortical tissue with a less dense network. Tumors develop an increased interstitial fluid pressure, compressing the thin-walled tumor vessels, which results in reduced circulation, explaining our findings [19]. Further, the haphazard network predisposes to a disorganized flow direction described in vivo [20].

The material was initially collected to study norepinephrine sensitivity in renal and tumor vessels [14]. The vessels of the specimens were maximally relaxed using papaverine, then perfused with increasing doses of norepinephrine. At each level, an injection of approximately <300,000 15 µm spheres, with different radioactive labeling, was done. Thus, capillary blocking from previously injected 15 µm spheres could prevent contrast filling. However, there are 0.8 to 1 million glomeruli in a human kidney and the afferent arteriolus splits into 6 to 8 loops. Thus, 5 to 8 million passages are present. At maximum, approximately 3 × 300,000 spheres were injected, i.e., a minority of passages were blocked. After injection of the microspheres, no increase in perfusion pressure despite constant flow was found, which indicates no significant vascular blocking. Since the manual contrast infusion time and pressure were not standardized, this could add to the varying degrees of contrast filling. The incomplete contrast filling is likely due to the viscosity of the suspension; however, it has a rheology similar to blood with a normal hematocrit of 0.45 [15]. All three differently labeled spheres will inevitably add to the autoradiogram blackening. The initial injection at vascular relaxation was done with I^125^ which has a low energy gamma emission (35 keV) with a higher absorption in the film than the higher energies from Ce^141^ and Ru^103^ used in the following injections. Thus, the autoradiograms represent essentially the perfusion at maximal vascular relaxation.

In 1896, a postmortem kidney was first injected with a suspension of red lead particles [21]. The infusion of micronized Barium sulphate [22] has a drawback of higher viscosity than liquid iodine-containing contrast media, necessitating the relatively high infusion pressure. However, it will result in high X-ray absorption and high contrast. Further, it will not diffuse from the vascular volume, resulting in good definition. The tumor vessels permit protein leakage, but extravascular micronized particles could not be observed. More confluent lakes of contrast could be observed, probably representing ruptured sinusoids. Liquid contrast media prevents further material handling, i.e., fixation and histological preparation, which would eliminate the contrast medium. The Barium sulphate will withstand handling, but as stated above, immunohistochemistry will, to some degree, rinse the contrast from the 4 µm sections. Therefore, we studied the vascular contrast in paraffin sections using dark-field microscopy before immunohistochemistry. This procedure allowed analysis of the fraction of vessels being contrast-filled by relocalizing the same area after CD31 immunostaining, which showed that only a fraction of the CD31-positive vessels were contrast-filled.

Some glomeruli totally or partially lacked contrast, as did most peritubular cortical capillaries. These findings agree with the micro-CT results where only the vascular network, including the glomeruli, are seen, but not the efferent and peritubular capillaries. In some sagittal angiograms and autoradiograms, it is clear that parts of the cortex are neither contrast-filled nor perfused with microspheres. This finding probably represents an unperfused auxiliary renal artery or occlusion caused by tumor growth nearby. Mixing of microspheres is crucial for representative distribution. In our perfusion setup the spheres were injected into the tubing to the renal artery. A low variation between different cortical samples within each specimen indicated sufficient mixing.

Unexpectedly, a minority of vasa recta were contrast-filled. Since few peritubular capillaries were contrast-filled, the filled vasa recta likely originated from juxtamedullary glomeruli, where the efferent arterioli pass directly into the medulla. Gross vascularity was visualized via contrast infusion but was also seen in more detail after immunohistochemical staining of the CD31 epitope. We chose CD31 as endothelial marker rather than CD34, since CD31 also labels immature vessels, whereas CD34 is more selective towards mature vasculature [23,24]. We used darkfield microscopy for visualization of the contrast resulting in intense light scatter, as shown above, and for enhanced visualization of the denatured immunoprotein precipitates at IHC [17]. The low-angle illumination principle was also used to visualize entire microslides with sections 0.5–1 cm^2^, giving a better overview of the vascular network. As seen in illustrations of vascularity, including X-ray angiograms, micro-CT and darkfield microscopy, the contrast produced is high. Presumably, a lower concentration of Barium sulphate with lower viscosity would provide sufficient contrast. A lower perfusion pressure would be needed and possibly a better vascular filling would result.

## 5. Conclusions

We demonstrate the utility of a set of techniques to characterize the complex vascular system of renal and tumor tissue both on a structural and functional level. The techniques apply well to other tissues and organs suitable for perfusion studies. Furthermore, the findings show that clear cell renal cell carcinoma is well vascularized on a structural level but functionally lacks perfusion. We speculate that this might curtail access to the cancer by therapeutic agents and require further study.

## Figures and Tables

**Figure 1 cancers-16-01939-f001:**
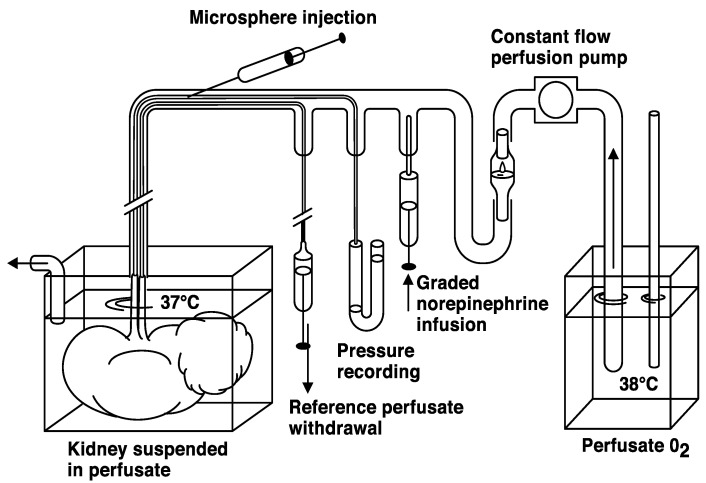
Schematic representation of the perfusion system used for the study.

**Figure 2 cancers-16-01939-f002:**
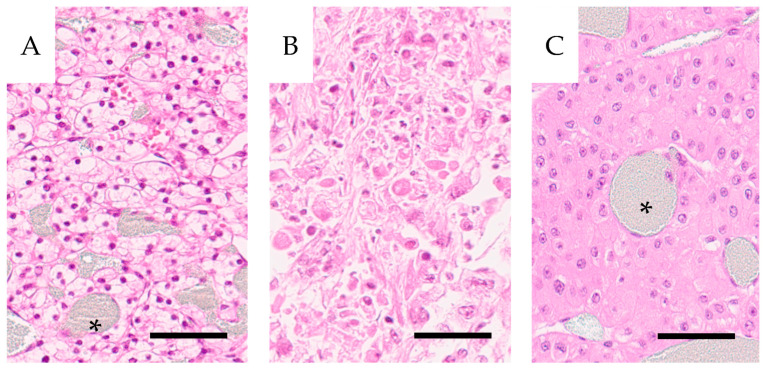
Histology of three cases included in the study. (**A**) Well-differentiated clear cell carcinoma (H7.1) PRU relative to cortex 1.3; (**B**) high-grade clear cell carcinoma with rhabdoid features and eosinophilic cytoplasm (H4.6) PRU relative to cortex 24; (**C**) an oncocytoma (H14), PRU relative to cortex 41. Barium sulphate contrast is visible as a grayish hue filling the tumor vasculature (asterisk). All cases were stained with hematoxylin-eosin. Scale bar = 100 µm.

**Figure 3 cancers-16-01939-f003:**
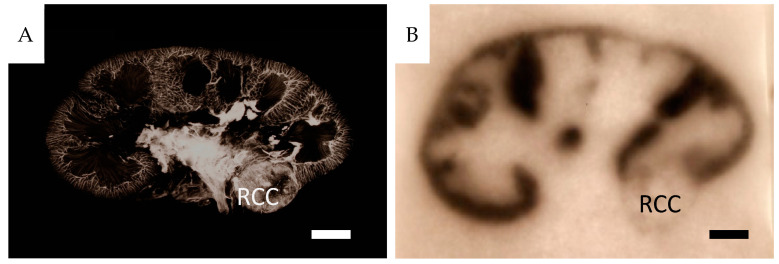
The vascular tree and perfusion properties of a small renal cell carcinoma. (**A**) Barium-angiogram of a sagittal section of a clear cell renal cell carcinoma specimen (H25), with the characteristic renal vascular pattern of the kidney. A well-vascularized tumor is also seen (RCC). (**B**) Autoradiogram of the adjacent tissue section visualizing the very poor nutritive <15 µm capillary perfusion of the tumor. The whole specimen PRU is 0.57. Scale bar = 1 cm. RCC: renal cell carcinoma.

**Figure 4 cancers-16-01939-f004:**
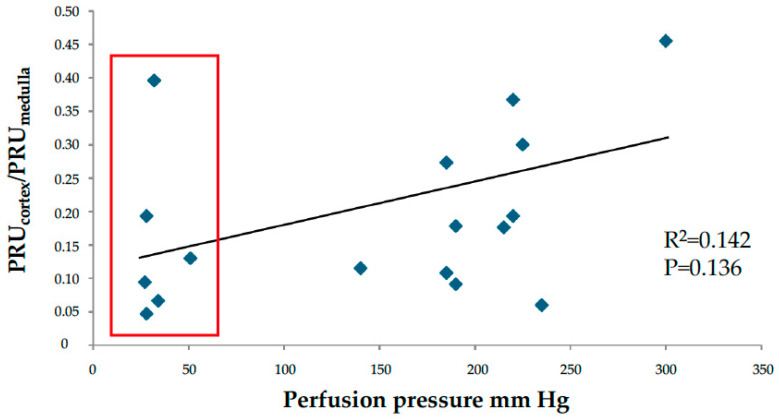
The fraction PRU in cortex vs. medulla plotted against the contrast perfusion pressure. The red rectangle shows the cases that were not contrast perfused at the end of the experiments.

**Figure 5 cancers-16-01939-f005:**
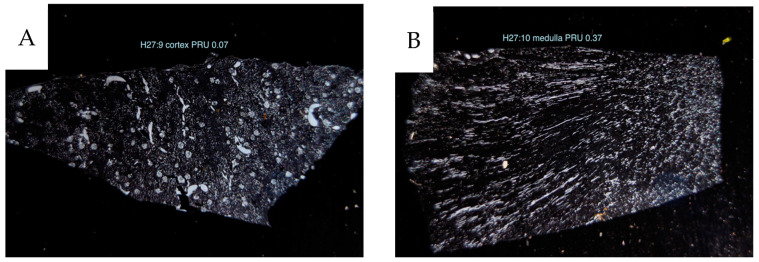
Dark-field macro photographs of 4 µm sections of kidney tissue. (**A**) cortical tissue, PRU 0.07. (**B**) medullary tissue, “PRU” 0.37, B (H27).

**Figure 6 cancers-16-01939-f006:**
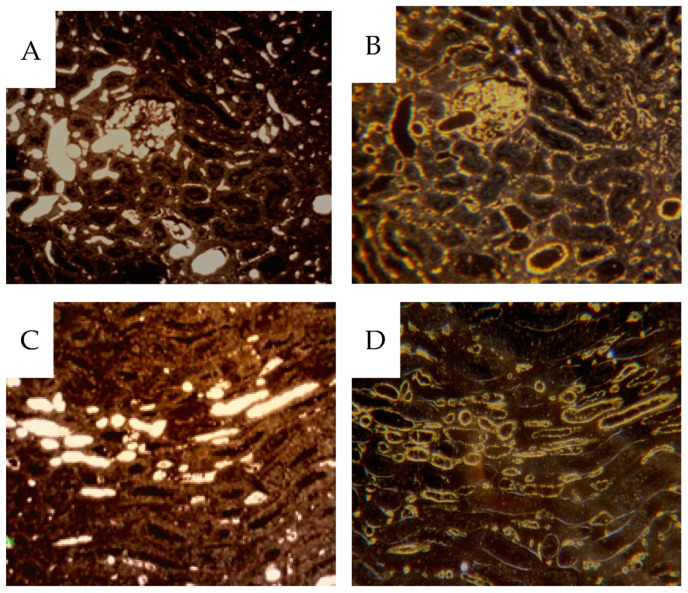
Darkfield microscopy of 4 µm sections from cortical tissue (**A**,**B**) and from medullary tissue (**C**,**D**) (H28). Contrast-filled vessels are seen in the unstained sections (**A**,**C**), while the same area in the same section is seen after CD31 IHC (**B**,**D**), where most contrast has been rinsed during processing. Only a fraction of CD31 positive vessels have been contrast perfused.

**Figure 7 cancers-16-01939-f007:**
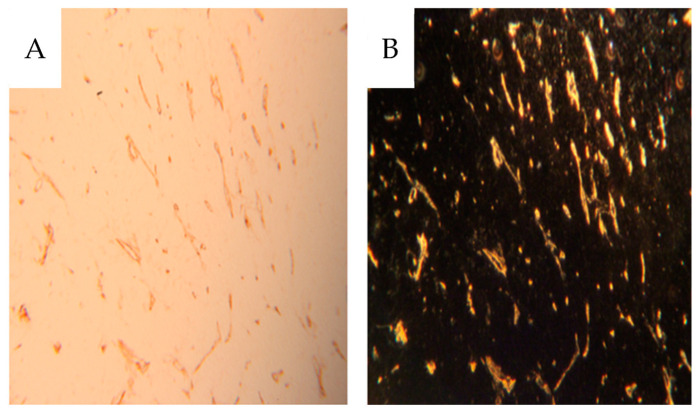
Immunohistochemical staining for CD31 in a section of tumor tissue as seen using brightfield (**A**) and dark-field (**B**) microscopy. There is no contrast in this specimen.

**Figure 8 cancers-16-01939-f008:**
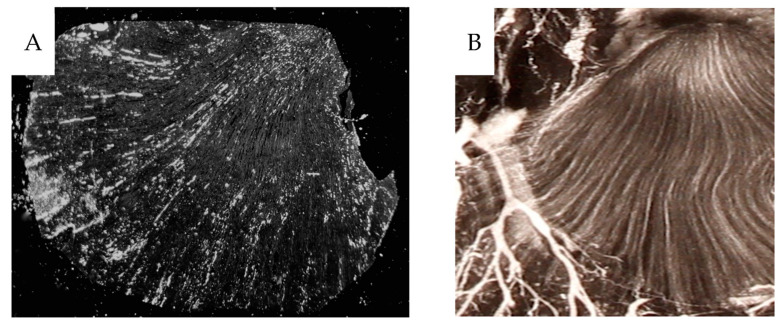
(**A**) The partially contrast filled vasa recta converging towards the apex are seen in the darkfield macrophotograph of a 4 µm section. (**B**) A similar pattern is seen on a contrast angiogram of a 1 mm thick sagittal section.

**Figure 9 cancers-16-01939-f009:**
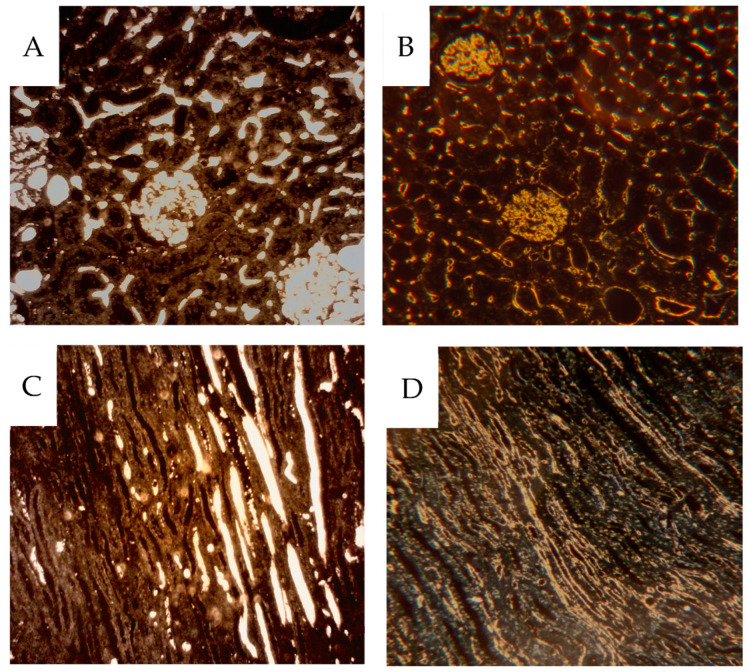
Darkfield microscopy of cortical tissue; (**A**) a contrast perfused specimen with dilated peritubular vessels compared to (**B**) a CD31 stained, non-contrast perfused specimen; (**C**) medullary tissue from a contrast-perfused specimen with widened vasa recta compared to (**D**) a CD31-stained specimen not contrast-injected.

**Figure 10 cancers-16-01939-f010:**
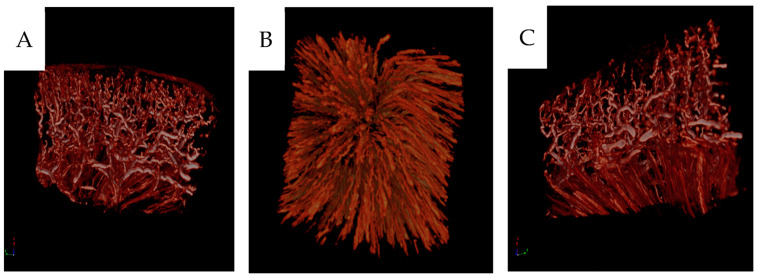
(**A**) Three-dimensional micro-CT reconstruction of renal cortical vessels with glomeruli and juxtacortical medulla. (**B**) Vasa recta in the medulla as seen from the periphery towards the pyramid’s apex. (**C**) The corticomedullary zone.

**Figure 11 cancers-16-01939-f011:**
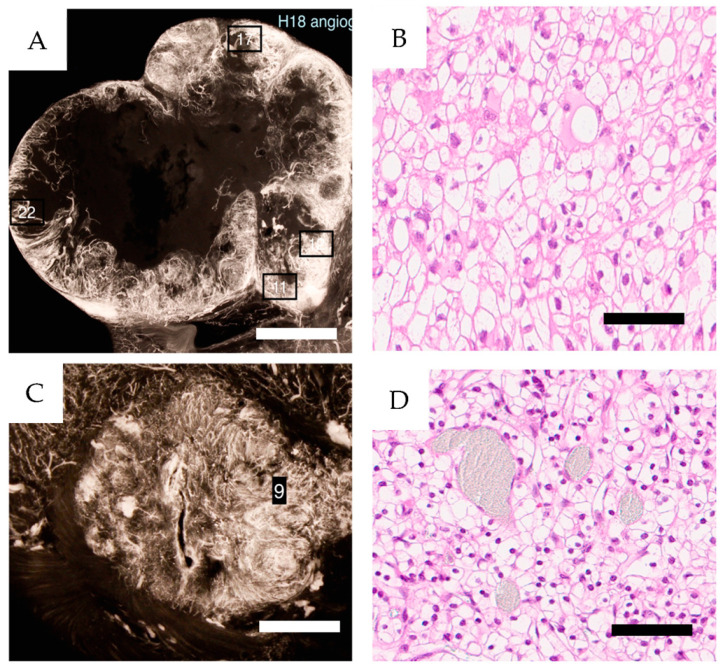
(**A**,**C**) angiograms from CCRCC case H18 (ISUP grade 2) and H11 (ISUP grade 1), PRU relative to cortex 20 and 3 respectively. To the right, Htx-Eosin sections (20×) from areas 22 and 9 as indicated in the angiograms (**B**,**D**). Areas 11, 12 and 17 displayed identical morphology. Scale bars 2 cm in (**A**), 5 mm in (**C**) and 100 µm in (**B**,**D**).

**Figure 12 cancers-16-01939-f012:**
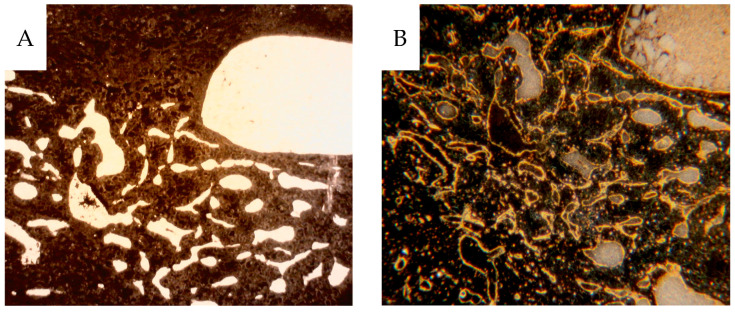
(**A**) Contrast-filled vessels as seen prior to CD31 staining. (**B**) The same tissue section following IHC for CD31. (H27:15, ISUP 2, PRU relative to cortex 14).

**Figure 13 cancers-16-01939-f013:**
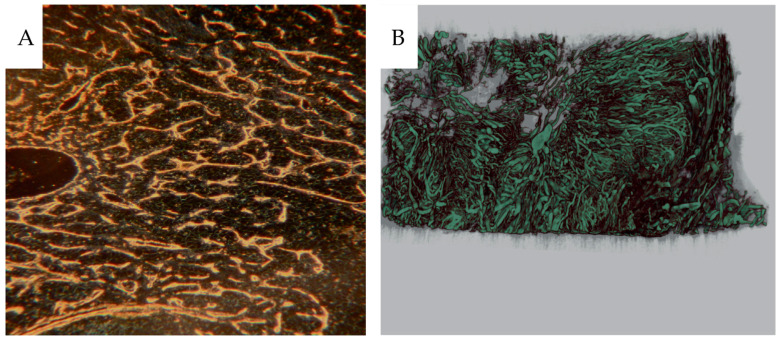
(**A**) CD31-stained tumor section of CCRCC from a non-contrast-infused specimen probably due to compressed vessels. (H2:7, ISUP 2, PRU relative to cortex 1.5). (**B**) 3D micro-CT reconstruction of the tumor vasculature with radiating vessels towards a necrotic area (H14-4, oncocytoma, PRU relative to cortex 14).

**Table 1 cancers-16-01939-t001:** Survival data on the clear cell renal carcinomas according to ISUP grade, survival and cause of death. Cases in italics indicate death of renal cancer, whereas standard font indicates death from other causes.

Survival				
ISUP grade		<5 years	5–10 years	>10 years
	1	H3, H7, H10, H16, H17	H11	H8, H9, H29
	2	H2, H15	H6, H13, H22, H24, H26	H18, H20, H28
	3	H5, H12	H23	H19
	4	H4, H21		H27

## Data Availability

The data presented in this study are available in this article and Appendix A.

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
