# Peer review of "Ex Vivo Vascular Imaging and Perfusion Studies of Normal Kidney and Tumor Vasculature"

_cancers, 2024, doi:10.3390/cancers16101939_

Round 1

Reviewer 1 Report

Comments and Suggestions for Authors

The manuscript "Ex-vivo Vascular Imaging and Perfusion Studies of Normal  Kidney and Tumor Vasculature" by Hultborn is devoted to studying the vascular structure of normal and tumorous kidneys. The work has a historical angle, combining modern and older techniques.

As such, the work can be of interest to a broad audience. It can be published if deficiencies are addressed.

1)      It isn't easy to read the article. It consists of huge paragraphs. For example, the whole Discussion (3 pages) contains only one paragraph. The article needs to be structured better, with logically connected paragraphs.

2)      Line 361: "Microscopy of the thin 4 μm sections disclosed few 15 μm spheres." How can you be sure that 15um spheres were not removed during slicing into 4um pieces? At least, it requires an explanation or references.

3)      Line 244: peripheral resistance has units. It is fine if you use the PRU abbreviation for them; however, it must be stated.

4)      Line 258: "115 samples obtained from 22 specimens, the PRU could be determined." It is unclear how the peripheral resistance of samples was determined.

Minor deficiencies:

5)      There is some awkward formatting. Lines 98-99: Same, lines 11-118. 259-260, 331-332

6)      Line 196: "During the study period" What was the study period? It was mentioned in Line 357. However, it needs to be mentioned in Methods.

7)      Line 204: Abbreviations like ISUP and CCRCC must be defined on their first appearance. DMBA (line 379)

8)      Line 385: "the into" the noun is missing

9)      Lines 409-410: "Cerium141  and Ruthenium103." The authors used unusual notation. I would recommend using element symbols instead.

10)   Line 415: "Further, it will not diffuse from the vascular volume, which also results in good definition." What about leaky angiogenic vessels? You used submicron particles.

11)   Line 418: "Bariumsulphate" should be split into two words on several occasions (e.g., line 448)

Comments on the Quality of English Language

Moderate editing is required. A lot of strange grammatical constructs

Author Response

Dear Reviewer! Please see the attachment. Best Regards Martin Johansson

Reviewer 2 Report

Comments and Suggestions for Authors

1. It would be helpful for the readers if the estimates of the continuous variables such as mean perfusion pressure, mean perfusion flow are provided in median (range) rather than mean (SD).

2. Can sub-group analysis be done based on the tumor types (clear cell carcinomas ISUP grade 1, ten tumors as ISUP grade 2, 4 204 tumors as CCRCC of eosinophilic type, ISUP grade 3, and 3 tumors as CCRCC of sarco- matoid/rhabdomyoid type, ISUP grade 4)?

3. Please provide the r2 and p-value in the Figure 4 for the regression line of best fit.  

4. In the statistical analysis section, please describe how were the data represented, statistical tests used, threshold p-value for significance, software used, etc.

Author Response

(The authors gave the same response as above.)

Reviewer 3 Report

Comments and Suggestions for Authors

1.What is the basic takeaway message of this research study is not very clear. The manuscript adds incremental value to the knowledge base in understanding the kidney vasculature and the renal tumor environment. Kidneys post nephroectomy were studied with micronized barium perfusion to demonstrate the vascular architecture and the radioactive microspheres of diameter <15 μm were used to determine the PRU (peripheral resistance unit). Similarly renal tumors were examined following the same procedure. Differential presence of renal vasculature was observed in the renal architecture with reduced vasculature and increased resistance in the medulla compared to the cortex. Following this line of thought renal tumors also provided data showing increased resistance in the center of the tumor compared to the peripherally expressed increased vasculature; this indicated the lack of nutritive flow to the center of the tumor. Many solid tumors including the renal indication have been previously reported to have dedicated anaerobic environment-dependent signaling mechanism to improvise unique vasculature-independent nutrient supply via ion transport mechanisms involving TRP channel proteins to deal with the issues of nutrient supply for tumor growth and proliferation. The vascular tumors are relatively common in the skin and soft tissue but their prevalence is extremely rare in the kidney (Omiyale AO. Primary vascular tumours of the kidney. World J Clin Oncol. 2021 Dec 24;12(12):1157-1168. doi: 10.5306/wjco.v12.i12.1157. PMID: 35070735; PMCID: PMC8716994.). Due to this fact it is not discussed very well how the knowledge gathered in the current study would help understanding renal cell carcinoma.

2. Discussion section (lines 356-359): Wrongly described information on existence of IHC and micro CT in clinical practice and medical research. The principle of IHC has existed since the 1930s, but it was not until 1941 that the first IHC study was reported.(Ref: Duraiyan J, Govindarajan R, Kaliyappan K, Palanisamy M. Applications of immunohistochemistry. J Pharm Bioallied Sci. 2012 Aug;4(Suppl 2):S307-9. doi: 10.4103/0975-7406.100281. PMID: 23066277; PMCID: PMC3467869.). Microcomputed tomography (Micro-CT) was developed in 1972. Ref: (https://imaging.rigaku.com/learning/micro-ct#:~:text=Micro%2DCT%3A%20Brief%20history,-The%20best%2Dknown&text=It%20was%20developed%20in%201972,goes%20back%20another%2055%20years.) and became a popular technique employed in medical and research use in 1980 when 3 million CT scan examinations had been recorded. (Ref: https://catalinaimaging.com/history-ct-scan/#:~:text=By%20the%20year%201973%2C%20the,scan%20examinations%20had%20been%20recorded.)

Author Response

(The authors gave the same response as above.)

Round 2

Reviewer 3 Report

Comments and Suggestions for Authors

I am satisfied with the responses provided by the authors addressing my comments and concerns. The revised manuscript appears much improved.